# New e:b-Friedo-Hopane Type Triterpenoids from *Euphorbia peplus* with Simiarendiol Possessing Significant Cytostatic Activity against HeLa Cells by Induction of Apoptosis and S/G2 Cell Cycle Arrest

**DOI:** 10.3390/molecules24173106

**Published:** 2019-08-27

**Authors:** Jin-Hai Yu, Dong-Xiang Wu, Zhi-Pu Yu, Yu-Peng Li, Yin-Yin Wang, Shu-Juan Yu, Hua Zhang

**Affiliations:** School of Biological Science and Technology, University of Jinan, 336 West Road of Nan Xinzhuang, Jinan 250022, China

**Keywords:** *Euphorbia peplus*, triterpenoids, simiarendiol, ECD calculation, apoptosis, cell cycle arrest

## Abstract

Seven rare e:b-friedo-hopane-type triterpenoids including four new (**1**–**4**) and three known (**5**–**7**) ones with **5** being first reported as a natural product, together with five other known triterpenoids (**8**–**12**), were isolated from the nonpolar fractions of the ethanolic extract of *Euphorbia peplus*. Structural assignments for these compounds were based on spectroscopic analyses and quantum chemical computation method. The structural variations for the C-21 isopropyl group, including dehydrogenation (**1** and **3**) and hydroxylation at C-22 (simiarendiol, **2**), were the first cases among e:b-friedo-hopane-type triterpenoids. Simiarendiol (**2**) bearing a 22-OH showed significant cytostatic activity against HeLa and A549 human tumor cell lines with IC_50_ values of 3.93 ± 0.10 and 7.90 ± 0.31 μM, respectively. The DAPI staining and flow cytometric analysis revealed that simiarendiol (**2**) effectively induced cell apoptosis and arrested cell cycle at the S/G2 phases in a dose-dependent manner in HeLa cells.

## 1. Introduction

The plant of *Euphorbia peplus* Linn. (Euphorbiaceae) is an annual herb native to Europe and North Africa and is nowadays naturalized to Asia, America, and Australia [1]. The whole plant is a well-known herbal medicine and has been used for the treatment of various skin diseases, such as sunspots, warts, and corns [2]. Previous phytochemical studies on this plant have mostly focused on its content in diterpenoids as jatrophane, ingenane, pepluane, and segetane types, which is of interest due to their diverse structures and broad spectrum of biological activities [3,4,5]. Of these diterpenoids, ingenol mebutate have been approved in the United States and the European Union in 2012, for treatment of actinic keratosis [6]. Additionally, in the nonpolar extractions of *E. peplus*, four triterpenoids, lanosterol, cycloartenol, 24-methylenecycloartanol, and simiarenol were also reported [4,7], among which simiarenol belongs to a very rare class of e:b-friedo-hopane-type triterpenoids.

In our continuing research to explore bioactive natural products from plant resources and also to extend the knowledge towards the aforementioned rare triterpenoid molecules, the whole plants of *E. peplus* were collected from the Panlong district of Kunming, China, and the lipid fractions of the ethanol extract were carefully investigated. As a result, 12 triterpenoids (Figure 1) were separated and structurally characterized. Seven (**1**–**7**) of these compounds incorporate an unusual e:b-friedo-hopane skeleton, of which compounds **1**–**4** were previously unreported and compound **5** with a rare diene group across A/B rings was first obtained as a natural product. The structural variations on the isopropyl residue at C-21, such as dehydrogenation in **1** and **3** and hydroxylation at C-22 in **2**, were reported for the time among e:b-friedo-hopane-type triterpenoids. Most interestingly, compared with other analogues, the presence of the 22-OH group in **2** significantly increased the cytotoxicity against a panel of tested human tumor cell lines (HeLa, A549, MCF-7, and MDA-MB-231, Appendix A). Further biological evaluations demonstrated that simiarendiol (**2**) inhibited the proliferation of HeLa cells through induction of apoptosis and cell cycle arrest at S/G2 phases in a dose-dependent manner. Herein, the isolation, structure elucidation, and biological assessments for these compounds are presented.

## 2. Results and Discussion

Compound **1** was obtained as a white amorphous powder. It displayed a proton adduct ion peak at *m/z* 425.3789 ([M + H]^+^, calcd 425.3778) in the (+)-HRESIMS spectrum, which along with the ^13^C NMR data uncovered a molecular formula of C_30_H_48_O, corresponding to seven indices of hydrogen deficiency. In the ^1^H NMR spectrum, there were three proton signals observed at the downfield region, of which two singlet resonances at *δ*_H_ 4.87 and 4.67 (Table 1) suggested the presence of a terminal double bond, while the one at *δ*_H_ 5.62 (Table 1) indicated the existence of a trisubstituted double bond. This was further confirmed by the presence of four olefinic carbon resonances at *δ*_C_ 146.2 (C), 142.2 (C), 122.1 (CH), and 111.3 (CH_2_) (Table 2) in the ^13^C and DEPT^135^ spectra, which also displayed 26 additional aliphatic carbon signals including seven methyls, nine methylenes, five methines (one oxygenated), and five quaternary carbons. Two double bonds accounted for two out of the seven indices of hydrogen deficiency, and this required five rings in the framework of Compound **1**. The aforementioned information suggested that Compound **1** was a pentacyclic triterpenoid, whose e:b-friedo-hopane skeleton, a rare skeleton bearing much variations in B and E rings as compared to the hopane-type triterpemoids [8], was finally established by examination of the 2D NMR data (Figure 2). In detail, inspection of the ^1^H–^1^H COSY spectrum led to the establishment of five spin–spin coupling systems (**a**–**e**), as drawn in bold bonds (Figure 2A). Subsequently, seven sets of HMBC correlations (Figure 2A) of H_3_-23(24)/C-3 (*δ*_C_ 76.5), C-4 (*δ*_C_ 41.0), and C-5 (*δ*_C_ 142.2); H-6/C-5 and C-10 (*δ*_C_ 50.4); H_3_-25/C-8 (*δ*_C_ 44.4), C-9 (*δ*_C_ 35.0), C-10, and C-11 (*δ*_C_ 34.3); H_3_-26/C-8, C-13 (*δ*_C_ 38.9), C-14 (*δ*_C_ 39.7), and C-15 (*δ*_C_ 29.3); H_3_-27/C-12 (*δ*_C_ 29.0), C-13, C-14, and C-18 (*δ*_C_ 51.9); H_3_-28/C-16 (*δ*_C_ 34.9), C-17 (*δ*_C_ 43.3), C-18, and C-21 (58.9); and H_3_-29/C-21, C-22 (*δ*_C_ 146.2), and C-30 (*δ*_C_ 111.3), allowed the aforementioned five fragments to be connected to each other via seven quaternary carbons and, thus, assembled the gross structure, as shown. In addition, the HMBC correlations from H_3_-23(24) to C-5 and from H-6 to C-5 and C-10 located one double bond at Δ^5^, while those from H_3_-29 to C-22 and C-30 positioned the other one at Δ^22(30)^. Furthermore, the molecular component revealed the presence of one hydroxyl group, which was attached to C-3, as confirmed by its downfield chemical shift (*δ*_C_ 76.5). It was worth noting that the presence of the double bond at Δ^22(30)^ in **1** was the first case in the reported e:b-friedo-hopane-type triterpenoids.

The relative configuration of Compound **1** was assigned mainly by analysis of the ROESY spectrum (Figure 2B). First, H-2α (*δ*_H_ 1.87), H-10, H-15α (*δ*_H_ 1.38), H-19α (*δ*_H_ 1.33), Me-23, Me-27, and Me-28 were assigned to be α-oriented and on the same side of the molecule by the ROESY correlations of H_3_-23/H-2α and H-10; H-10/H_3_-27; H_3_-27/H-15α and H-19α; and H-15α/H_3_-28. It was followed by the assignment of β-orientation for H-1β (*δ*_H_ 1.59), H-7β (*δ*_H_ 1.95), H-12β (*δ*_H_ 1.52), H-16β (*δ*_H_ 1.72), H-18, H-21, Me-25, and Me-26, as determined by the cross-peaks of H-1β/H_3_-25; H_3_-25/H-7β and H-12β; H-7β/H_3_-26; H_3_-26/H-16β and H-18; and H-18/H-21. Particularly, the strong ROESY correlations of H_3_-23 with H-2α and H-10 supported that they were axially bonded and ring A adopted a chair-like conformation (see Figure 2B), which together with the coupling pattern of H-3 (broad singlet, Table 1) confirmed that the 3-OH group was also axially bonded and thus β-directed. As shown in Figure 3A, the calculated ECD curve for Compound **1** highly matched its experimental one, thus establishing its absolute configuration, as displayed. Therefore, Compound **1** was elucidated as e:b-friedo-hop-5,22(30)-dien-3β-ol.

Compound **2** was assigned a molecular formula of C_30_H_50_O_2_ as inferred from its (+)-HRESIMS ion peak at *m/z* 425.3774 ([M + H − H_2_O]^+^, calcd 425.3778) and the ^13C NMR data. Comparison of the 1^H and ^13^C NMR data (Table 1 and Table 2) with those of Compound **1** revealed that Compound **2** also incorporated the e:b-friedo-hopane skeleton and differed from Compound **1** in the C-21 side chain, where the isopropenyl group in Compound **1** was replaced by an isopropanol-2-yl group in Compound **2**. Such conclusion was further confirmed by analysis of the HMBC spectrum (Appendix A), where the correlations from H_3_-29(30) to C-21 (*δ*_C_ 61.2) and C-22 (*δ*_C_ 73.4) were evident. The relative configuration of Compound **2** was assigned to be the same as that of Compound **1** by their highly similar NMR data and identical ROESY correlations (Appendix A). As shown in Figure 3B, the ECD spectrum of Compound **2** was in good agreement with that of Compound **1**, which then established the same absolute configuration for Compound **2**. The hydroxylation at C-22 in Compound **2** was also reported for the first time in the e:b-friedo-hopane family. The structure of Compound **2** was thus characterized as e:b-friedo-hop-5-en-3β,22-diol. In addition, Compound **2** showed significant cytotoxic activity in the following bioactive test and was, therefore, used to performer a series of mechanism studies and we named it ‘simiarendiol’ according to its analogue simiarenol (**6**).

Compound **3** had a molecular formula of C_30_H_44_O_2_ which was deduced from the (+)-HRESIMS ion peak at *m*/*z* 437.3396 ([M + H]^+^, calcd 437.3414) and the ^13^C NMR data. Its UV spectrum showed a strong absorption peak at 245 nm, which was indicative of the presence of an α,β-unsaturated carbonyl group. As with Compound **1**, two downfield singlet signals (*δ*_H_ 4.88, 4.67) in the ^1^H NMR data (Table 1) disclosed the existence of a terminal double bond. Compared to those of Compound **1**, the NMR data (Table 1 and Table 2) of Compound **3** revealed high structural similarities between the two co-metabolites, implying that Compound **3** was also a e:b-friedo-hopane-type triterpenoid. Comprehensive analyses of the 2D NMR spectra (Appendix A) finally established the planar structure of Compound **3**. In particular, the key HMBC correlations from H_3_-23(24) to C-3 (*δ*_C_ 213.7), and H_2_-7 and H-8 to C-6 (*δ*_C_ 199.5), located two keto carbonyls at C-3 and C-6, respectively, while those from H_3_-23(24) to C-5 (*δ*_C_ 136.3) and H_3_-25 to C-10 (*δ*_C_ 166.0) fixed the tetrasubstituted double bond at Δ^5(10)^. In addition, the terminal double bond was also positioned at Δ^22(30)^ like that of Compound **1**, as conformed by the key HMBC correlations from H_3_-29 to C-21 (*δ*_C_ 58.8), C-22 (*δ*_C_ 145.8), and C-30 (*δ*_C_ 111.6), and H_3_-28 to C-21. The relative configuration of Compound **3** was assigned to be the same as that of Compound **1** (except C-10) by analysis of the ROESY spectrum (Appendix A), with the absolute configuration being established by comparing the experimental ECD curve with the calculated one (Figure 4A). Compound **3** was thus elucidated, as depicted.

Compound **4** displayed a proton adduct ion peak at *m*/*z* 439.3562 ([M + H]^+^, calcd 439.3571) in the (+)-HRESIMS spectrum, supportive of a molecular formula of C_30_H_46_O_2_ in agreement with the ^13^C NMR data. Analyses of the ^1^H and ^13^C NMR data of Compound **4** (Table 1 and Table 2) revealed that most signals were comparable to those of Compound **3**, with the exception occurring at the C-21 side chain, where an isopropyl group in Compound **4** was observed to replace the isopropenyl group in Compound **3**. Further analyses of the 2D NMR spectra (Appendix A) confirmed the aforementioned conclusion. The absolute configuration of Compound **4** was determined to be the same as that of Compound **3**, based on the observation of their nearly overlapping ECD curves (Figure 4). Therefore, the structure of Compound **4** was fully characterized.

Compound **5**, featuring a rare diene group across the A/B rings, was initially reported as a chemosynthetic derivative in the structural elucidation of simiarenol (**6**) in 1966 [8], but any NMR data for this compound was not provided so far. It was first reported as a natural product in the current work, and the assignment of NMR data in pyridine-*d*_5_ (Table 1 and Table 2) was also accomplished by analyses of the 2D NMR spectra (Appendix A). In addition, we also determined its absolute configuration as shown by using the TD-DFT based ECD calculation (Figure 5).

By comparing the NMR data with those reported in the literature, the other known analogues were identified as simiarenol (**6**) [9], simiarenone (**7**) [10], 24-methylenecycoartanol (**8**) [11], cycloartenol (**9**) [12], lanosta-7,24-dien-3*β*-ol (**11**) [13], lanosta-8,24-dien-3*β*-ol (**11**) [14], and lupeol (**12**) [15].

All of the isolates were subjected to cell viability assessment toward four human cancer cell lines (HeLa, A549, MCF-7, and MDA-MB-231) by the MTT method. As a result, simiarendiol (**2**) showed moderate to significant activity against these four cell lines with IC_50_ values ranging from 3.93 to 14.22 μM (Table 3), whereas the others were inactive (Appendix A). It was easily concluded that the presence of the 22-OH group was crucial for the cytotoxicity of Compound **2**.

As simiarendiol (**2**) exhibited the best activity toward HeLa cell line (IC_50_ = 3.93 ± 0.10) and inhibited the cell viability in a dose-dependent manner, a preliminary mechanistic investigation of Compound **2** was carried out in the HeLa cells. DAPI staining and fluorescence microscope were first employed to detect the cell morphological changes upon treatment with different doses of Compound **2**. As shown in Figure 6, compared with the control group, morphological changes including cell shrinkage, pyknosis, and karyorrhexis were observed, especially at 6.0 and 12.0 μM concentration. The DAPI staining experiment demonstrated that Compound **2** effectively induced apoptosis of the HeLa cells. A subsequent flow cytometric Annexin V/PI double staining assay was conducted to quantify the apoptosis. As shown in Figure 7, the percentages of total apoptosis (early and late apoptosis) for control (0.1% DMSO) and Compound **2** at 3.0 μM were 7.98% and 9.83%, respectively. With the concentrations of Compound **2** increasing to 6.0 and 12.0 μM, the percentages of cell apoptosis increased remarkably to 27.27% and 49.94%, respectively. The results revealed that Compound **2** induced a significant apoptosis of the HeLa cells in a dose-dependent manner.

We also explored whether simiarendiol (**2**) led to a cell cycle arrest in the HeLa cells by flow cytometry. As shown in Figure 8, Compound **2** clearly arrested the S and G2 phases of the cell cycle, as the percentages of cells at S/G2 phases increased from 6.84%/23.17% (control) to 19.36%/35.08% (6.0 μM) and 28.14%/38.36% (12.0 μM), respectively. Such results demonstrated that Compound **2** inhibited the proliferation of the HeLa cells via the induction of S/G2 phases arrest in a dose-dependent manner.

## 3. Experimental Section

### 3.1. General

Optical rotations were measured on a Rudolph VI polarimeter (Rudolph Research Analytical, Hackettstown, NJ, USA) with a 10 cm length cell. ECD and UV spectra were recorded on a Chirascan Spectrometer (Applied Photophysics Ltd., Leatherhead, UK) with a 0.1 cm pathway cell. NMR experiments were performed on a Bruker Avance DRX600 spectrometer (Bruker BioSpin AG, Fallanden, Switzerland) and referenced to the residual solvent peaks (CDCl_3_: *δ*_H_ 7.26, *δ*_C_ 77.16; pyridine-*d*_5_: *δ*_H_ 7.22, 7.58 and 8.74, *δ*_C_ 123.87, 135.91, and 150.35). ESIMS analyses were carried out on an Agilent 1260-6460 Triple Quad LC–MS instrument (Agilent Technologies Inc., Waldbronn, Germany). HR–ESIMS spectra were obtained on an Agilent 6545 Q-TOF mass spectrometer (Agilent Technologies Inc., Waldbronn, Germany). HPLC separations were performed using an Agilent 1260 series LC instrument (Agilent Technologies Inc., Waldbronn, Germany) equipped with an Agilent SB-C_18_ column (9.4 × 250 mm, Agilent Technologies Inc., Santa Clara, CA, USA). Column chromatography (CC) was performed on D101-macroporous absorption resin (Sinopharm Chemical Reagent Co. Ltd., Shanghai, China), Sephadex LH-20 (GE Healthcare Bio-Sciences AB, Uppsala, Sweden), and silica gel (300–400 mesh; Qingdao Marine Chemical Co. Ltd., Qingdao, China). All solvents used for CC were of analytical grade (Tianjin Fuyu Fine Chemical Co. Ltd., Tianjin, China) and solvents used for HPLC were of HPLC grade (Oceanpak Alexative Chemical Ltd., Goteborg, Sweden). Pre-coated silica gel GF_254_ plates (Qingdao Marine Chemical Co. Ltd., Qingdao, China) were used for thin-layer chromatography (TLC) analyses. All solvent mixtures used for analyses and separations (HPLC and CC) were presented in the ratio of volume-to-volume, unless otherwise specified.

### 3.2. Plant Material

The whole plants of *Euphorbia peplus* L. were purchased from Kunming Plantwise Biotech Co. Ltd. (Kunming, China) that selected the plants from the Panlong district of Kunming (25.03° N, 102.72° E) in July 2018. A voucher specimen was deposited at the School of Biological Science and Technology, University of Jinan (accession number: npmc-037).

### 3.3. Extraction and Isolation

The air-dried powder of *E. peplus* (10 kg) was extracted with 95% EtOH at room temperature for four times. After removal of the solvent under reduced pressure, the obtained residue (1.5 kg) was suspended in 2.0 L water and partitioned with EtOAc (3.0 L × 3). The EtOAc partition (470 g) was subjected to CC over D101-macroporous absorption resin (10.0 × 70 cm, 3.5 kg), eluted with EtOH-H_2_O (30%, 50%, 80%, and 95%), to afford four fractions (A, B, C, and D). Fraction D (190 g) was separated by silica gel CC (7.5 × 45 cm, 500 g), eluted with petroleum ether (PE)-EtOAc (20:1 to 5:1), to produce eight fractions (D1–D8). Fraction D2 (7.5 g) was separated on a silica gel CC (4.5 × 15 cm, 30 g), eluted with PE-EtOAc (30:1 to 10:1), to afford five subfractions, and the third one was then crystalized in methanol to yield Compound **7** (800 mg). Fraction D3 (7.8 g) was repeatedly crystalized in methanol to yield Compound **6** (700 mg). Fraction D4 (10.0 g) was chromatographed on a silica gel CC (4.5 × 18 cm, 36 g), eluted with PE-EtOAc (10:1 to 5:1), to generate two subfractions (D4-1 and D4-2). The first subfraction D4-1 was then separated by repeated silica gel CC and finally purified by HPLC (3.00 mL/min, 100% MeOH) to yield Compounds **12** (2.1 mg, *t*_R_ = 30 min), **10** (1.1 mg, *t*_R_ = 32.9 min), and **1** (1.3 mg, *t*_R_ = 33.8 min). Fraction D5 (6.5 g) was crystalized in methanol to give two subfractions D5-1 and D5-2. A portion of the crystallized part D5-1 was then purified by HPLC (3.00 mL/min, 100% MeOH) to yield Compounds **11** (3.5 mg, *t*_R_ = 30.2 min), **9** (10.3 mg, *t*_R_ = 31.6 min), and **8** (9.0 mg, *t*_R_ = 34.7 min). The mother solution D5-2 was separated on a silica gel CC (3.0 × 10 cm, 20 g) and eluted with PE-EtOAc (30:1 to 10:1) to afford two subfractions; the second fraction was then purified by HPLC (3.00 mL/min, 100% MeOH) to yield Compound **3** (0.9 mg, *t*_R_ = 14.0 min) and Compound **4** (4.5 mg, *t*_R_ = 17.7 min). The last fraction D8 (6.3 g) was subjected to Sephadex LH-20 CC (4.0 × 100 cm, 50 g, CHCl_3_-MeOH, 1:1) to return two subfractions, and the second one was then purified by HPLC (3.00 mL/min, 95%–100% MeOH-H_2_O) to afford Compound **2** (7.2 mg, *t*_R_ = 27.0 min) and Compound **5** (1.5 mg, *t*_R_ = 32.6 min).

e:b-friedo-Hop-5,22(30)-dien-3β-ol (Compound **1**, 98.5% in purity): white amorphous powder; [*α*]_D_^25^ +37.8 (*c* 0.27, CHCl_3_); ECD (*c* 0.04, MeCN) *λ* (Δ*ε*) 205 (+2.48) nm; ^1^H NMR data (CDCl_3_) (see Table 1) and ^13^C NMR data (CDCl_3_) (see Table 2; (+)-HR-ESIMS *m/z* 425.3789 [M + H]^+^ (calcd. for C_30_H_49_O, 425.3778).

e:b-friedo-Hop-5-en-3β,22-diol (simiarendiol, Compound **2**, 99.0% in purity): white amorphous powder; [*α*]_D_^25^ +54.4 (*c* 0.72, CHCl_3_); ECD (*c* 0.04, MeCN) *λ* (Δ*ε*) 210 (+2.94) nm; ^1^H NMR data (CDCl_3_) (see Table 1) and ^13^C NMR data (CDCl_3_) (see Table 2); (+)-ESIMS *m/z* 465.2 [M + Na]^+^; (+)-HR-ESIMS *m/z* 425.3774 [M + H − H_2_O]^+^ (calcd. for C_30_H_49_O, 425.3778).

e:b-friedo-Hop-5(10),22(30)-dien-3,6-dione (Compound **3**, 96.4% in purity): white amorphous powder; [*α*]_D_^25^ −47.9 (*c* 0.09, CHCl_3_); UV (MeCN) *λ*_max_ (log *ε*), 245 (4.84) nm; ECD (*c* 0.04, MeOH) *λ* (Δ*ε*) 213 (−5.66), 253 (+1.41), 297 (−1.35), 349 (−0.59) nm; ^1^H NMR data (CDCl_3_) (see Table 1) and ^13^C NMR data (CDCl_3_) (see Table 2); (+)-ESIMS *m/z* 459.2 [M + Na]^+^; (+)-HR-ESIMS *m/z* 437.3396 [M + H]^+^ (calcd. for C_30_H_45_O_2_, 437.3414).

e:b-friedo-Hop-5(10)-en-3,6-dione (Compound **4**, 98.6% in purity): white amorphous powder; [*α*]_D_^25^ −37.6 (*c* 0.12, CHCl_3_); UV (MeCN) *λ*_max_ (log *ε*), 246 (4.61) nm; ECD (*c* 0.04, MeOH) *λ* (Δ*ε*) 209 (−4.82), 253 (+0.89), 298 (−0.88), 352 (−0.39) nm; ^1^H NMR data (CDCl_3_) see Table 1 and ^13^C NMR data (CDCl_3_) see Table 2; (+)-ESIMS *m/z* 461.2 [M + Na]^+^; (+)-HR-ESIMS *m/z* 439.3562 [M + H]^+^ (calcd. for C_30_H_47_O_2_, 439.3571).

e:b-friedo-Hop-1(10),5-dien-3β-ol (Compound **5**, 96.4% in purity): white amorphous powder; [*α*]D^25^ +50.8 (*c* 0.08, CHCl_3_); UV (MeCN) *λ*_max_ (log *ε*) 236 (4.55) nm; ECD (*c* 0.04, MeCN) *λ* (Δ*ε*) 237 (+3.07) nm; ^1^H NMR data (CDCl_3_) see Table 1 and ^13^C NMR data (CDCl_3_) see Table 2; (+)-ESIMS *m/z* 425.2 [M + H]^+^; (+)-HR-ESIMS *m/z* 425.3777 [M + H]^+^ (calcd. for C_30_H_49_O, 425.3778).

### 3.4. ECD Calculations

The ChemDraw_Pro_14.1 software with an MM2 force field was used to establish the initial conformations of the target molecules. Conformational searches using mixed torsional/Low-mode sampling method with MMFFs force field in an energy window of 3.01 kcal/mol were carried out by means of the conformational search module in the Maestro 10.2 software (Maestro Technologies, Inc., Trenton, NJ, USA). The re-optimization and the following TD–DFT calculations of the re-optimized conformations were all performed with the Gaussian 09 software (Gaussian, Inc., Wallingford, CT, USA) [16] at the B3LYP/6-311G(d,p) level, in vacuo. Frequency analysis was performed as well to confirm that the re-optimized conformers were at the energy minima. Finally, the SpecDis 1.64 software (https://specdis-software.jimdo.com/) [17] was used to obtain the Boltzmann-averaged ECD spectra.

### 3.5. Cell Viability Assay

The cell viability assay of Compounds **1**–**12** toward four human cancer cell lines [MCF-7 & MDA-MB-231 (breast cancer), HeLa (cervical cancer), and A549 (lung cancer)] (National Infrastructure of Cell Line Resource, Beijing, China) was tested using the MTT method, as described formerly in [18]. Doxorubicin was used as the positive control.

### 3.6. DAPI Staining

HeLa cells were plated in confocal dishes for 24 h and then incubated with DMSO and simiarendiol (Compound **2**) at 3.0, 6.0, and 12.0 μM for 48 h. After incubation, the cells were settled in 4% paraformaldehyde for 10 min and washed three times with PBS. Finally, the cells were incubated with DAPI (Beyotime Biotechnology, Shanghai, China) for 15 min in the dark, washed three times again, and then photographed by a Leica DMi8 fluorescence microscope (Leica Microsystems, Wetzlar, Germany).

### 3.7. Flow Cytometric Analysis of Cell Apoptosis

Cell apoptotic analysis was performed with an Annexin–V/FITC apoptosis kit (BD Biosciences, New York, NY, USA). Briefly, HeLa cells were seeded in a 6-well plate at 2 × 10^5^ cells/well for 24 h and then treated with simiarendiol (Compound **2**) at the concentrations of 3.0, 6.0, and 12.0 μM for 48 h. According to the manufacturer’s instruction, the cells were harvested, washed twice with PBS, and gently re-suspended in binding buffer. Finally, the cells were incubated with Annexin–V/FITC (5 μL) and PI (5 μL) in the dark, at room temperature, for 15 min. A total of 30,000 events were collected for each sample and analyzed by a flow cytometer (ACEA Biosciences, San Diego, CA, USA), and the percentage of apoptotic cells was calculated by the NovoExpress analysis software (ACEA Biosciences, San Diego, CA, USA).

### 3.8. Cell Cycle Analysis

HeLa cells were seeded in a 6-well plate at 2 × 10^5^ cells/well and incubated at 37 °C for 24 h. Then, the cells were respectively treated with 3.0, 6.0, and 12.0 μM simiarendiol (Compound **2**) for 36 h. After incubation, the cells were harvested by centrifugation, washed twice with PBS, and fixed in 70% ice-cold ethanol overnight at 4 °C. After removal of the ethanol, the cells were re-suspended with ice-cold wash buffer and then centrifuged. The obtained cells were stained with PI staining solution (Genview, El Monte, CA, USA) and kept in dark place at 37 °C for 30 min. Finally, cell cycle analysis was performed on a flow cytometer (ACEA Biosciences, San Diego, CA,, USA), and the proportion of cells at various stages were analyzed by NovoExpress analysis software.

## 4. Conclusions

In this work, the chemical fractionation of the nonpolar constituents from *E. peplus* has resulted in the isolation and identification of seven rare e:b-friedo-hopane-type triterpenoids, including four new ones (Compound **1**–**4**) and three known derivatives (Compound **5**–**7**), as well as five other known triterpenoid co-metabolites (Compound **8**–**12**). Structural modifications of the isopropyl residue at C-21 in Compounds **1**–**3** were first observed for the e:b-friedo-hopane triterpenoids, thus, enriching the chemical diversity of this rare class of natural products. Our initial bioactive evaluation revealed that the presence of 22-OH in Compound **2** was responsible for its significant cytotoxicity compared with the other analogues. Further experiments including DAPI staining and flow-cytometric analysis demonstrated that Compound **2** effectively induced cell apoptosis and arrested the cell cycle at S/G2 phases in the HeLa cells, in a dose-dependent manner. Overall, the current studies demonstrated the possibility of further structural modifications on e:b-friedo-hopane triterpenoids for better cytostatic agents that could be developed into potential antitumor leads. Chemical transformation of the abundant isolates, such as **6** and **7**, into 22-hydroxylated analogues and more derivatives remains a work in due course.

## Figures and Tables

**Figure 1 molecules-24-03106-f001:**
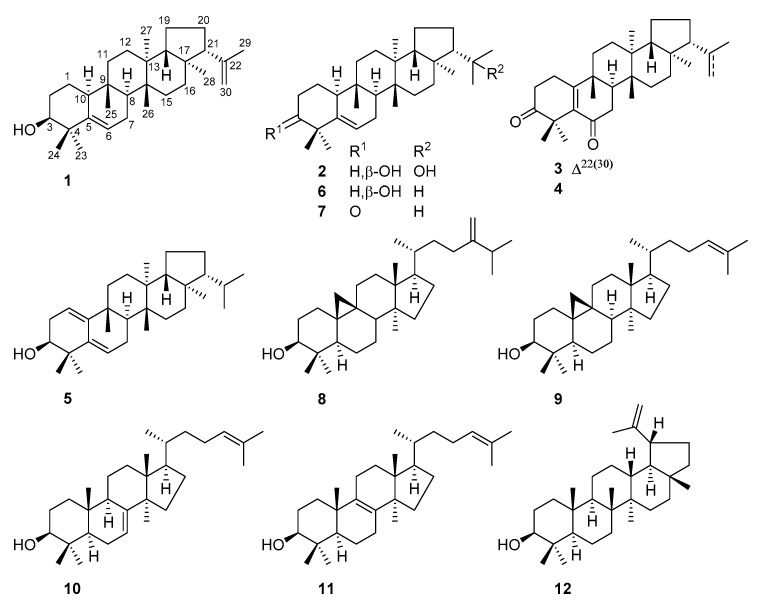
Structures of Compounds **1**–**12**.

**Figure 2 molecules-24-03106-f002:**
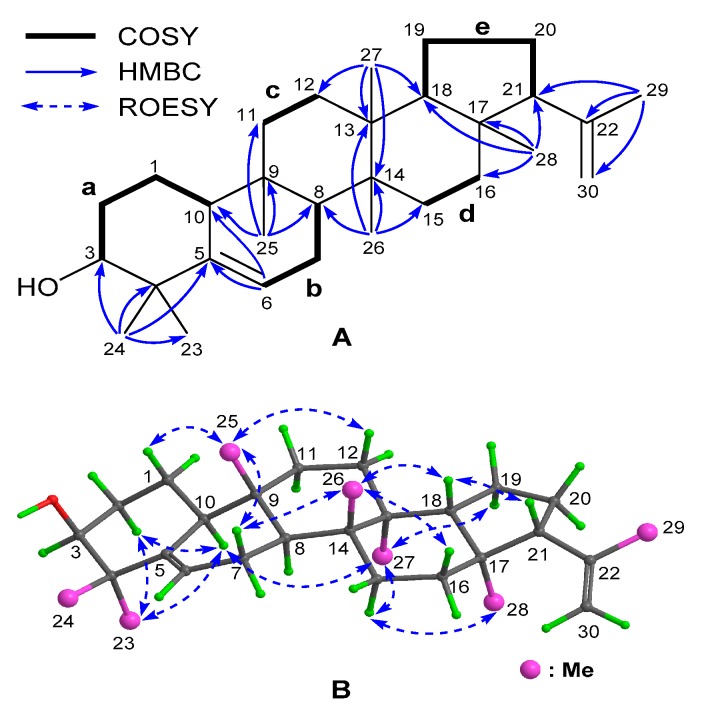
^1^H–^1^H COSY and Key HMBC Correlations for Compound **1** (**A**); Key ROESY Correlations for Compound **1** (**B**).

**Figure 3 molecules-24-03106-f003:**
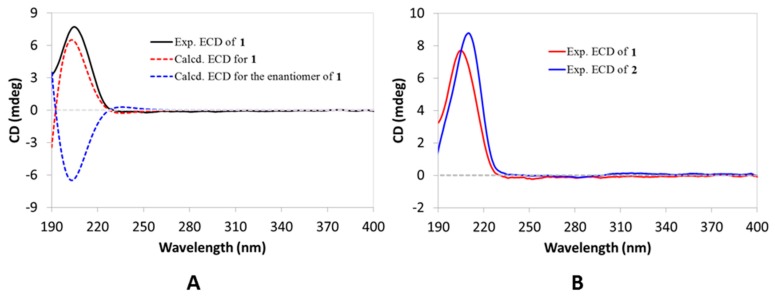
(**A**) Experimental ECD spectrum of Compound **1** (black) compared with the calculated ECD spectra of **1** (red) and its enantiomer (blue). (**B**) Experimental ECD spectra of Compound **1** (red) and Compound **2** (blue).

**Figure 4 molecules-24-03106-f004:**
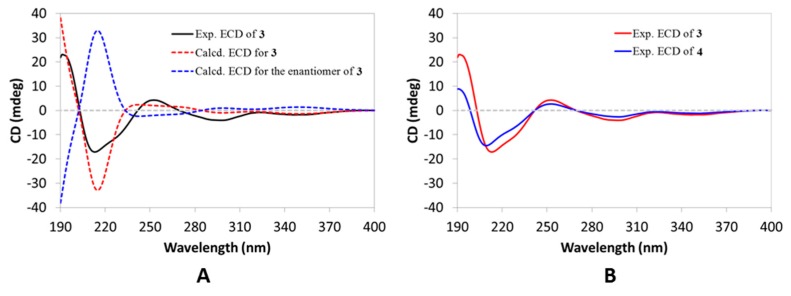
(**A**) Experimental ECD spectrum of Compound **3** (black) compared with the calculated ECD spectra of Compound **3** (red) and its enantiomer (blue). (**B**) Experimental ECD spectra of Compound **3** (red) and Compound **4** (blue).

**Figure 5 molecules-24-03106-f005:**
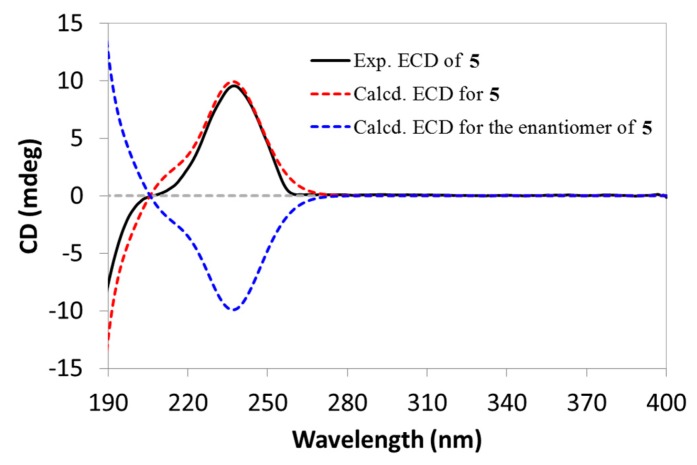
Experimental ECD spectrum Compound **5** (black) compared with the calculated ECD spectra of Compound **5** (red) and its enantiomer (blue).

**Figure 6 molecules-24-03106-f006:**
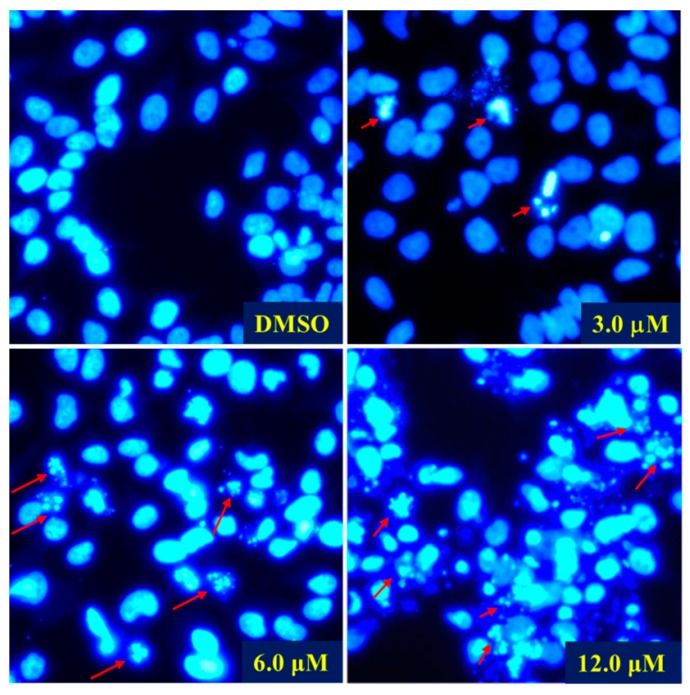
Morphological observation for Hela cells. HeLa cells were treated with vehicle control (DMSO) and Compound **2** at different concentrations for 48 h, stained with DAPI and viewed under a fluorescence microscope at a magnification rate of 200. Red arrows represent the altered nuclear morphology in a dose-dependent manner.

**Figure 7 molecules-24-03106-f007:**
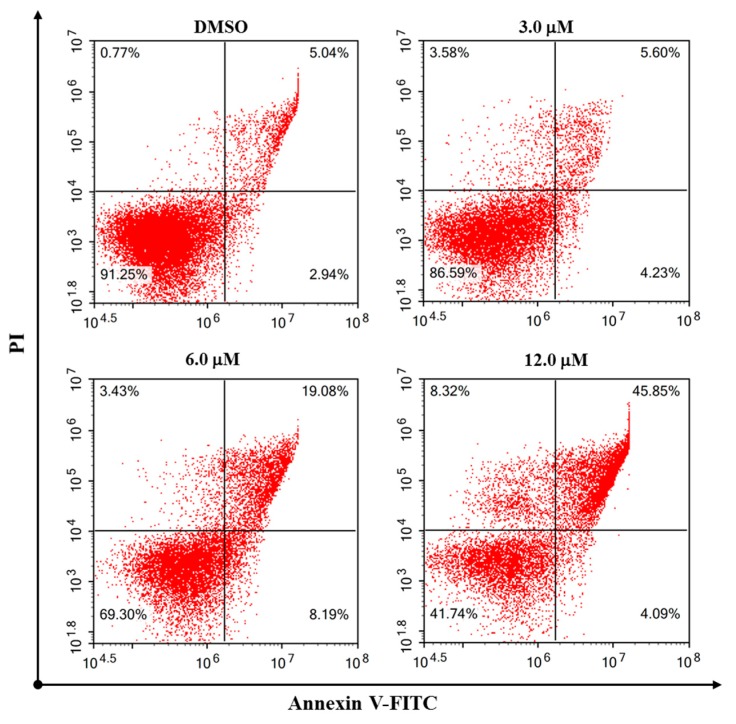
Compound **2** induced cell apoptosis in the Hela cells. Hela cells were treated with the vehicle control (DMSO) and Compound **2** for 48 h, at different concentrations, stained with Annexin V and Propidium Iodide (PI), and were detected by flow cytometry.

**Figure 8 molecules-24-03106-f008:**
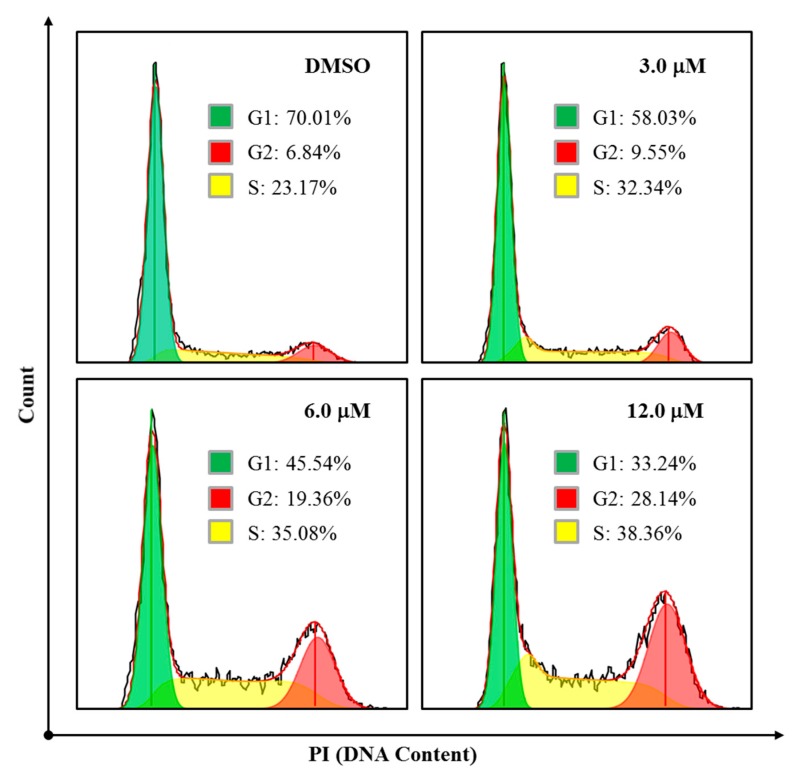
Compound **2** caused cell cycle arrest in the HeLa cells. The HeLa cells were treated with the vehicle control (DMSO) and Compound **2** for 36 h, at different concentrations, stained with PI, and then finally analyzed by flow cytometry.

**Table 1 molecules-24-03106-t001:** ^1^H NMR spectroscopic data for Compounds **1**–**5**.

No.	1 *^a^*	2 *^a^*	3 *^a^*	4 *^a^*	5 *^b^*
1α 1β	1.49, m 1.59, m	1.49, m 1.58, m	2.57, m 2.70, m	2.57, m 2.70, m	5.47, dd (5.6, 2.6)
2α 2β	1.87, m 1.69, m	1.86, m 1.69, m	2.66, m 2.43, m	2.66, m 2.43, m	2.73, ddd (17.8, 5.6, 5.6) 2.45, ddd (17.8, 10.1, 2.6)
3	3.47, br s	3.47, br s			3.88, ddd (10.1, 5.6, 5.3)
6	5.62, br d (5.7)	5.61, dt (6.1, 2.1)			5.79, t (4.1)
7α 7β	1.83, m 1.95, m	1.82, m 1.94, m	2.34, dd (17.9, 4.3) 2.42, dd (17.9, 13.9)	2.34, dd (17.9, 4.3) 2.41, dd (17.9, 13.9)	2.11, m, 2H
8	1.52, m	1.49, m	2.11, dd (13.9, 4.3)	2.09, dd (13.9, 4.3)	1.81, dd (9.9, 6.7)
10	2.08, m	2.08, m			
11α 11β	1.65, m 1.51, m	1.64, m 1.50, m	1.87, td (13.3, 5.3) 1.55, m	1.87, m 1.55, m	2.17, td (13.7, 5.7) 1.47, m
12α 12β	1.09, m 1.52, m	1.08, m 1.53, m	1.18, m 1.62, td (13.4, 4.6)	1.19, m 1.63, m	1.13, m 1.56, m
15α 15β	1.38, td (13.1, 4.2) 1.24, m	1.42, m 1.21, dt (13.6, 3.5)	1.43, td (13.2, 4.2) 1.20, m	1.43, m 1.22, m	1.14, m, 2H
16α 16β	1.50, m 1.72, m	1.70, m, 2H	1.56, m 1.74, m	1.64, m 1.63, m	1.53, m 1.58, m
18	1.76, dd (12.7, 7.4)	1.59, m	1.74, m	1.58, m	1.58, m
19α 19β	1.33, m 1.45, m	α 1.32, m β 1.44, m	1.46, m 1.35, m	1.37, m 1.23, m	1.38, m 1.25, m
20α 20β	1.72, m 1.68, m	1.75, m 1.72, m	1.73, m 1.68, m	1.85, m 1.00, m	1.13, m 1.77, m
21	2.06, m	1.46, m	2.06, t (9.8)	0.95, m	0.93, m
22				1.44, m	1.38, m
23	1.05, s	1.04, s	1.47, s	1.47, s	1.49, s
24	1.14, s	1.14, s	1.33, s	1.33, s	1.19, s
25	0.90, s	0.90, s	1.18, s	1.17, s	1.09, s
26	1.04, s	1.01, s	1.04, s	1.01, s	1.05, s
27	0.93, s	0.92, s	0.95, s	0.95, s	0.90, s
28	0.69, s	0.99, s	0.69, s	0.79, s	0.73, s
29	1.73, s	1.31, s	1.73, s	0.89, d (6.5)	0.85, d (6.5)
30	4.87, s 4.67, s	1.17, s	4.88, s 4.68, s	0.83, d (6.5)	0.90, d (6.5)
3-OH					6.20, d (5.3)

*^a^* In CDCl_3_; *^b^* In pyridine-*d_5_*.

**Table 2 molecules-24-03106-t002:** ^13^C NMR spectroscopic data for Compounds **1**–**5**.

No.	1 *^a^*	2 *^a^*	3 *^a^*	4 *^a^*	5 *^b^*
1	18.2	18.2	25.9	25.9	114.6
2	27.9	27.9	35.4	35.5	33.7
3	76.5	76.5	213.7	213.7	72.9
4	41.0	41.0	46.7	46.7	39.4
5	142.2	142.2	136.3	136.3	142.1
6	122.1	122.1	199.5	199.6	120.9
7	24.2	24.2	36.8	36.8	24.9
8	44.4	44.3	42.8	42.8	42.5
9	35.0	34.9	40.3	40.0	36.8
10	50.4	50.4	166.0	166.0	147.6
11	34.3	34.2	30.1	30.1	31.9
12	29.0	29.2	28.4	28.4	29.5
13	38.9	38.7	38.7	38.6	40.2
14	39.7	39.5	40.1	40.0	40.7
15	29.3	29.1	28.6	28.6	29.5
16	34.9	36.0	34.7	35.4	36.0
17	43.3	44.0	43.2	42.9	43.2
18	51.9	51.8	51.7	51.7	52.4
19	20.0	19.8	19.9	20.0	20.5
20	25.1	22.8	25.1	28.5	29.0
21	58.9	61.2	58.8	60.2	60.4
22	146.2	73.4	145.8	30.9	31.3
23	29.2	29.2	26.6	26.6	24.3
24	25.6	25.6	21.9	21.9	20.4
25	18.0	18.1	21.8	21.8	25.1
26	16.0	15.9	15.9	15.8	16.2
27	14.9	15.3	15.2	15.4	15.5
28	17.0	17.3	17.2	16.5	16.7
29	25.0	29.6	24.9	22.1	23.4
30	111.3	32.0	111.6	23.0	22.4

*^a^* In CDCl_3_; *^b^* In pyridine-*d_5_*.

**Table 3 molecules-24-03106-t003:** Cytotoxicity of Compound **2** against four human tumor cell lines.

Compounds	Human Cell Lines (IC_50_ = mean ± SD, μM)			
	HeLa	A549	MCF-7	MDA-MB-231
Compound **2**	3.93 ± 0.10	7.90 ± 0.31	10.51 ± 1.21	14.22 ± 1.63
doxorubicin	0.91 ± 0.11	0.73 ± 0.06	0.61 ± 0.04	0.63 ± 0.08

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
