# Peer review of "New e:b-Friedo-Hopane Type Triterpenoids from Euphorbia peplus with Simiarendiol Possessing Significant Cytostatic Activity against HeLa Cells by Induction of Apoptosis and S/G2 Cell Cycle Arrest"

_molecules, 2019, doi:10.3390/molecules24173106_

Round 1

Reviewer 1 Report

Dear scientists

Publication is interesting and based on extensive research workshop. However, authors should improve certain parts:

fig. 6 - too low magnification makes it impossible to assess cell morphology please indicate the source of the origin of the cell lines chater 3,3 - please complete the description of the reagents if possible, please change too old literature /8, 14/

Author Response

Point 1: Fig. 6 -too low magnification makes it impossible to assess cell morphology.

Response to Point 1: The pictures in Fig. 6 have been magnified as suggested.

Point 2: please indicate the source of the origin of the cell lines

Response to Point 2: The origin of the cell lines have been added in chapter 3.5 Cell Viability Assay.

Point 3: chater 3.3 - please complete the description of the reagents

Response to Point 3: All reagents have been described in chapter 3.1 general. 

Point 4: if possible, please change too old literature 8, 14

Response to Point 4: Literatures 8 and 14 are too important to be changed.

Reviewer 2 Report

Generally speaking, the arrows of HMBC denote correlations between the proton and the carbon. In Fig.2, the arrows show the correlations between carbon and carbon. It is better to correct them.

The configurations of protons were determined based on ROESY. Some of them can be proved based on 3J coupling constants.

Does the 3D structure determined based on ROESY shown in Fig.2 agree with the 3D structure obtained from the molecular modeling?

Author Response

Point 1: Generally speaking, the arrows of HMBC denote correlations between the proton and the carbon. In Fig.2, the arrows show the correlations between carbon and carbon. It is better to correct them.

Response to point 1:  In natural product chemistry, the arrows that denote HMBC correlations were usually presented as we depicted. So it is not necessary to change them. 

Point 2:  The configurations of protons were determined based on ROESY. Some of them can be proved based on 3J coupling constants.

Response to point 2:  Yes, the 3J coupling constants can be able to determine some relative configurations in some cases, but it must be based on the well-established preferential conformations.  On the contrary, the ROESY correlations are proved to be direct proofs to establish the relative configurations. So, when the ROESY correlations are evident, it is not necessary to discuss the  3J coupling constants.

Point 3: Does the 3D structure determined based on ROESY shown in Fig.2 agree with the 3D structure obtained from the molecular modeling?

Response to point 3: Yes, they are agreement.

Reviewer 3 Report

Dear Authors

Research methodology is modern and developed. The work should be completed in several points.

line 217 - GPS coordinates, chapter 3,3 - all reagents should be described /producer, country/, where the cell lines MB 231, HeLa, A 549 came from? all references should have DOI, headings 8 and 14 appear to be outdated

Author Response

Point 1:  line 217 -GPS coordinates

Response to point 1: The detailed GPS coordinates have been added as suggested.

Point 2:  chapter 3,3 -all reagents should be described/producer, country,where the cell lines MB 231, HeLa, A 549 came from?

Response to point 2: All reagents have been described in chaper 3.1 general. The origin of the cell lines have been added in chapter 3.5 Cell Viability Assay.

Point 3:  all references should have DOI, headings 8 and 14 appear to be outdated

Response to point 3: The references that have DOI have been added DOI address as highlighted in yellow. In addition, literatures 8 and 14 are too important to be changed.